# Does Gender Impact Technology Adoption in Dual-Purpose Cattle in Mexico?

**DOI:** 10.3390/ani12223194

**Published:** 2022-11-18

**Authors:** Oriana Villarroel-Molina, Carmen De-Pablos-Heredero, Cecilio Barba, Jaime Rangel, Anton García

**Affiliations:** 1Animal Science Department, Rabanales University Campus, University of Cordoba, 14071 Cordoba, Spain; 2Department of Business Administration, ESIC University, Avda. Valdenigrales, s/n, Pozuelo de Alarcón, 28032 Madrid, Spain; 3Department of Business Economics (Administration, Management and Organization), Applied Economics II and Fundamentals of Economic Analysis, Rey Juan Carlos University, Paseo de los Artilleros s/n, 28032 Madrid, Spain; 4Mexico’s National Institute for Forestry, Agriculture and Livestock Research (INIFAP), Medellín de Bravo 94277, Mexico

**Keywords:** social network analysis, gender perspectives, dual-purpose cattle, technology adoption, centrality measures

## Abstract

**Simple Summary:**

Small-scale systems are the most widespread productive system in developing countries. However, these systems have historically suffered from a low technological level, which constitutes their main problem and endangers the maintenance of the system and rural people who depend for their livelihood on it. Therefore, deepening the knowledge of the technology adoption process is key to its improvement, as well as including a gender approach to develop more effective public policies. In this research, very small commercial farms with a similar structure and size in the Mexican tropics (n = 383) were analysed, and the Social Network Analysis (SNA) was used to identify technology adoption patterns among male and female farmers of dual-purpose cattle. Five technological packages were analysed and statistically significant differences were found between genders in the area of reproduction. The results have shown that women even when not occupying central positions, are close to the leaders in the network, quickly adopting the re-productive technologies the leaders selected. Furthermore, farms run by women were smaller although with greater specialization and milk productivity, increasing productivity by 20%.

**Abstract:**

This paper examines the role of women in the dual-purpose livestock system (DP) in Mexico through their technological adoption patterns and aims to evaluate whether there are significant differences between the technology adoption networks of men and women farmers. The sample was composed of 383 DP small farms with 20 or fewer cows and a high level of vulnerability. Social Network Analysis (SNA) was applied, and the centrality measures were calculated for the technological areas of management, feeding, genetics, reproduction, and animal health. Significant differences were found in reproduction levels between men and women farmers. Therefore, SNA was developed in this technological area where men mainly occupied central positions (brokers) while women were just close to the leaders in the network. The results have shown that farms run by women were smaller and presented higher levels of specialization and milk productivity (20% higher), and women prioritized those technologies linked to female reproductive efficiency. Moreover, women were deeply embedded in men’s networks through numerous ties and were capable of building connections with groups of farmers outside of their own group.

## 1. Introduction

The Sustainable Development Goals (SDGs) constitute a universal call to action to eradicate extreme poverty and protect the planet. The progress achieved since the SDGs expanded their scope to 17 goals has been substantial. By 2015, the world had already met the first goal of cutting global rates of extreme poverty and hunger by half. However, the scope of achievement has been uneven. Goal number five is dedicated exclusively to gender equality, since gender equality is not only a fundamental human right but also one of the essential tools to build a peaceful, prosperous and sustainable world [1,2]. On the other hand, gender equality and women’s empowerment also represent the goals and solutions of many other SDGs, such as Goal two, which seeks to end hunger and achieve food security in developing countries [3,4,5]. 

However, even though women are central to family food security and nutrition, being responsible for the food preparation and childcare in developing countries, challenges faced by women are more pronounced in the case of rural women. Furthermore, the gender perspective recognizes that some issues and constraints related to project success are gender-specific, and stem from the fact that men and women play different roles, have different needs, and face different constraints on several different levels.

According to data from the 2020 Population and Housing Census in Mexico, the second most populous country in Latin America, women represent 51.58% of the population. This proportion does not mean better personal and professional opportunities for the female gender in the country—quite the opposite, as Mexico faces social and economic inequalities and disadvantages in addition to high discrimination against women. This is evidenced by 77% of men being employed in primary, secondary, and tertiary economic activities compared to 45% of women [6,7,8]. Apart from this, the agricultural activity of Mexico supports 58% of the total production value and represents 42% of total income, where the activities developed by women are fundamental not only because they carry out diversified and traditional agricultural practices that maintain and improve crops, but also because they retain ancestral knowledge and traditional uses of the different crops and local varieties [8]. However, women still do not participate as equal partners in sustainable development practices, often have limited access to resources, and are excluded from decision-making processes [7]. Moreover, the female gender in the rural world has been historically invisible and discriminated against in all areas with unequal treatment [6]. Due to this, it is essential to promote the inclusion of women and to detect their needs, based on their experiences and perspectives, for sustainable technological and systemic change [6,7].

The dual-purpose livestock systems in the Mexican tropics respond to small-scale production, and are key to the food security of the inhabitants of the tropics; both in terms of provision and access to food, stability, and prices [8]. Globally, dual-purpose systems generate between 12% and 19% of the world’s milk and meat [9,10] and are often found on the poverty line, in fragile extensive systems with a high degree of marginalization. In addition, these subsistence farms showed a low level of technological innovation, which makes it difficult to access external inputs and makes them highly vulnerable to environmental disasters and economic turmoil [11]. In this context, the research problem was the gap in the current knowledge about the reasons for the low technology adoption rate in this livestock system and the grounds for why some farmers adopt or reject technologies potentially available to them. Therefore, a new approach is required to help improve this problem, as well as the full participation of women, but this will not happen until women are perceived as subjects of development [12]. In view of the above, there is a need to analyse the involvement of women in dual-purpose livestock production in Mexico.

Historically, low levels of technology adoption rate have been one of the main problems of small-scale livestock systems in Latin America and other developing countries [1,3,4,10,11,12], which negatively influences their productivity and threatens the food security of many families whose livelihoods depend on livestock farming [10,13]. The lack of technological innovation is due to multiple factors, such as low dimension, poor financial capability, lack of support for technological adoption, poor structures, risk aversion, and a misalignment between technological improvements and a farm’s objectives, amongst others [14,15,16]

Several methodologies have been used to assess the technological level: García et al. [17] and De-Pablos-Heredero et al. [10] assessed the impact of technological innovation on performance in dairy sheep farms in Spain; Rangel et al. [13] developed a technological characterization of dual-purpose livestock in Mexico; Cortes-Arriola et al. [18] and Cuevas-Reyes et al. [19] evaluated the influence of scale and location on the adoption of innovation, and Rada et al. [20] and Foster and Rosenzweigh [21] found an inverse relationship between a farm’s size and its land productivity. Furthermore, a growing body of literature applying social network analysis (SNA) to several fields of agricultural science and technological dissemination in agrolivestock systems has emerged [22,23,24,25,26,27,28,29,30,31,32,33], which has added a new dimension in understanding the role of social networks. Today, most studies using SNA have sought to understand naturally occurring network processes, and less attention has been paid to how social network characteristics and the use of SNA can be used to inform different adoption patterns according to gender, ethnic group, objectives, etc.

Currently, technological adoption is analysed through a holistic point of view, linking closely social factors such as producers’ characterization, their setting, and the type of relationship between themselves [31,34]. In addition, within social networks, factors such as technological learning, learning by doing and learning by imitation are of great importance, as well as adoption by observation among producers’ neighbours to reach an accumulated experience in a delayed manner [35]. However, low adoption rates are associated with the intrinsic uncertainty of innovation [36].

In this sense, technical advisors and local leaders play an important role in technological adoption and innovation diffusion, since SNA identifies diffusion patterns and units of analysis are frequently connected to an intensive network [31,37,38,39,40,41,42,43]. The improvement of technology adoption requires the deepest knowledge of how innovation spreads and what the relationships and links amongst stakeholders to transfer innovation are [38], and SNA use in smallholders facilitates the understanding of rural processes of innovation [23]. 

In this context, understanding gender roles is important for identifying effective support for local development, since men and women have different approaches to technological adoption in small-scale farms in developing countries, due to work division and cultural and social factors [6,12]. Furthermore, given the key role of women in agriculture, their inclusion and empowerment are fundamental requirements to achieve the well-being of rural communities. Gender analysis is a methodology that seeks to understand the distinct culturally and socially defined roles and tasks that women and men assume, both within the family and household system and within the community.

This study contributes to deepening the knowledge of the low technology adoption rate in DP livestock systems from the perspective of gender, as most of the studies in the field have estimated the network’s structure based on adoption intentions [6,11]. On the contrary, this research estimated networks from the reproductive technologies already adopted by farmers, identifying the actual technological adoption patterns by gender. According to De-Pablos-Heredero et al. [10] and Walther et al. [12], male and female farmers with different technological patterns and betweenness power in the network will achieve different productive results.

Therefore, an effective response requires knowing the skills and knowledge of women in livestock systems to identify solutions and improve the problem of low technology adoption in dual-purpose cattle systems. On the one hand, the gender perspective towards adoption as a key element in agricultural development processes has been slow, therefore concrete actions are still necessary to achieve tangible results. In addition, the role of women in rural areas of developing countries is not very visible, although it is key in the administration of generally complex households. The recognition of women’s work is the first step to face exclusion, although there is no clear perception of the roles that they play in rural areas. On the other hand, the predominant production system in rural Mexico is the family subsistence system, where the separation between agricultural activity and the family sphere is practically non-existent. Consequently, women’s work in the different fields is less visible than in other environments where the tasks are clearly delimited [44]. 

Since women are generally underrepresented in livestock studies, integrating the gender perspective into our technology adoption analysis contributes to understanding the main differences between men and women and how these influence technology adoption decisions. Our gendered analysis considers the diversity of social relationships that link farmers and measures the degree to which the structure of these relationships facilitates or limits technology adoption and productivity.

This paper aims to deepen the knowledge of the technological adoption process within the context of rural women, and dual-purpose cattle farms in Mexico to understand how gender impacts innovation processes. 

The paper draws on in-depth empirical research about women using technology in rural settings to explore how gender affects technology adoption.

The sequences of this research were: 1. Identify differences in technology adoption patterns between men and women. 2. Build a characterization of the technological and socio-economic indicators in the farms of dual-purpose cattle in Mexico by gender.

## 2. Materials and Methods

### 2.1. Study Area and Data Collection

This study was conducted in the central and south-eastern coastal regions of Mexico, and was complementary to previous work on dual-purpose technological innovation carried out by Rangel et al. [45], Rangel et al. [13], and Villarroel-Molina et al. [46,47]. The data of this research come from the research project “Adoption and evaluation of the impact of technology implemented in dual-purpose bovine systems of México”, approved by the Collegiate Group of the National Center for Disciplinary Research in Physiology and Animal Improvement of INIFIAP SIGI 21541832011. This project was approved on 14 March 2011 and was performed from 1 April 2013 to 30 April 2016. Data were collected in 2015 by direct surveys and updated in 2018.

The population was composed of 3285 commercial farms of DP receiving technical advice on feeding and economical management from the Livestock Technical Assistance Program (SAGARPA). A sample of 1275 farms was selected to build the typology of farms [13]. In this research, the group called “Dual Purpose smart farms” (between 10 and 19 cows) with 383 farmers was selected. Here, we focused on the more vulnerable and smaller group of commercial farmers with 20 or fewer cows, which represented 26% of the sample, distributed into dry and wet tropics. According to Rangel et al. [13], the 383 selected farms constituted a homogeneous cluster, small in size and with a low technological level which marketed most of the production. In addition, through the exploration of the data, we realized that the application of SNA is more significant and appropriate for groups with a lower technological level, since the differences can be better identified. Furthermore, being a group similar in economic dimension, the effect that economic factors may have on technology adoption disappears.

The main characteristics of this group are shown in Table 1. These producers exhibited an average age of 51.12 years old, the youngest farmer being 20 years of age, and the oldest 85 years of age, with nearly three people directly depending on the farms, generating 1.5 working units and 2.19 years of experience in technical advice. These were low- production and -dimension farms. The average size was 27.32 ha, 11.83 cows and 19.32 animal units (AU). The average annual production per farm per year was 11,267.24 L of milk, 245 kg of cheese, and 4.9 calves. The labour productivity was low (7511 L/W), and the same happens with the jobs created, with an average of 1.5 UW. Here, the unit of work (UW) was interpreted as a person who works on the farm more than 240 days per year or its equivalent in hours. Nevertheless, the annual capacity per cow and per hectare were 989.11 L/cow and 107.8 L/ha, respectively. In Rangel et al. (2020) [3,5,6,7], a broader description of the characteristics of this group of dual-purpose farmers can be found.

### 2.2. Technological Level in Dual-Purpose Farms

In previous work, 45 technologies of dual-purpose farms in the Mexican tropics were evaluated according to the methodological approach used by [17,45,48]. In Rangel et al. [13] the methodology used for the selection of innovations and its grouping into technological areas is detailed.

Briefly, the methodology used consists of two sequential steps [49]. In the first step, relevant technologies were identified by a panel of experts. In a second step, technologies were grouped into technological areas through multivariate analysis [50]. Forty-five technologies were distributed across five technology packages: eight technologies in the management area, fourteen in the feeding area, nine in genetics, seven in reproduction area, and seven in the animal health area. A technological innovation index was calculated for each technology and area, based on the proportion of technologies implemented in each area [48].

In small farms of dual-purpose cattle in Mexico, the technological level of adoption was low, only 47.0% of the technologies potentially available were used. Animal health area showed the higher technological level (higher than 72%), and management and genetics were the areas with medium technological adoption level (above 60%). On the contrary, the areas with the lowest use of technologies were feeding and reproductive (less than 30%). In the 383 farms, the technological levels within each group were quite homogeneous, therefore the use of technologies showed low dispersion of the results, with a coefficient of variation between 5 and 13% in each technological area (Table 2).

### 2.3. Statistical Analysis and SNA Methodology

SPSS Statistics 25.0 (Armonk, USA: IBM Corp.) for Windows software was used to perform the statistical analyses. Prior to the statistical analyses, the normality of the data distribution was evaluated using the Kolmogorov–Smirnov test (with the Lilliefors correction). For those variables that did not show a normal distribution, the Bartlett test was applied to assess if the data had equal variances. The parametric variables were compared using the analysis of variance (ANOVA) and the non-parametric variables through the Kruskal–Wallis test, establishing the gender as a fixed effect.

SNA was applied in small farms of dual-purpose cattle according to the following steps: In the first place, the centrality measures (Degree, Closeness, Eigenvector, Betweenness and Bonacich Power) were calculated with UCINET software for each technological area. The results obtained were subjected to a one-way ANOVA to assess whether there were statistically significant differences (*p* < 0.05) in the centrality measures between men and women. The focus was placed on those areas where significant differences by gender were found.

In those technological areas with significant differences (*p* < 0.05) in the centrality measures, SNA methodology was applied. To analyse the dual-purpose technology network, we constructed a two-mode network of 383 farmers and those technologies based on the technological package of each respondent. Two-mode networks, also known as affiliation networks [51,52,53], consist of recording instances in which individuals participate in or attend the same events, or where there are archival data indicating which people belong to which organizations [54,55]. After considering the different ways of analysing two-mode networks developed by [52,53,56,57], the two-mode data were transformed into a bipartite graph. For the aim of this paper, the bipartite graph was more appropriate to calculate centrality measures and offered significant results. The analysis and visualization of the dual-purpose cattle network in the Mexican tropics were carried out using the UCINET 6 for window (Freeman, Everett, and Borgatti; Harvard, USA) [58]. Subsequently, to visualize the two-mode network of farmers and technologies, we used Gower metrics scaling layout. Gower’s distance (or similarity) first computes distances between pairs of variables over two data sets and then combines those distances to a single value per record-pair. So, it is used to measure how different two records are. The nomenclature (fn) has been agreed to identify the farmers in the network.

Finally, a characterization of small farms of dual-purpose cattle by gender was completed. The mean structural, technical, economic, and technological indicators were calculated considering sex as fixed factor. The differences between groups were checked using a one-way ANOVA analysis.

## 3. Results

### 3.1. Differences in Centrality Measures between Men and Women

Table 3 shows the values of Mean, standard deviation (SD) and the coefficient of variation (CV) for the five technological packages studied: Management, Feeding, Genetics, Reproduction and Animal Health. The five technological areas showed similar values between the men and the women. In addition, no significant differences were found due to the gender factor (*p* > 0.05). Mean values and coefficients of variation were closer between both groups. Only reproductive area showed different variation coefficient values, 29.41% in the men compared to 19.75% in the women, although no significant differences were obtained.

In Table 4, the centrality measurement was shown (Degree, Betweenness, Closeness, Eigenvector and Bonacich Power). Centrality measures were calculated for five technological areas in the dual-purpose system in Mexico: feeding, reproduction, health, genetics, and management. A comparison of samples was performed using one-way ANOVA to identify whether there were statistically significant differences between men and women in these areas, and centrality measures descriptive statistics were calculated. The results showed that there were significant differences in the reproduction area for the values of degree, closeness, eigenvector, betweenness and Bonacich power (*p* < 0.05). Therefore, the rest of the SNA analysis was addressed in the reproduction area. The reproduction area technologies description is shown in Table 4. 

The centrality measures descriptive statistics for the group of men showed that, within the reproduction technological innovation network, the dual-purpose farmers had an average degree of 1.94, related to the number of technologies that these farmers have adopted. In the case of the women, the average degree was 1.38. Regarding the average betweenness, related to the ability to interfere with communication or information flowing through the network, it was 8.42 for the men, with a variation coefficient of 176.18%, while it was 4.59 for the women, with a variation coefficient of 223.14%. Closeness was the measure with the lowest coefficient of variation in reproduction (20.21), with an average of 837.34 in the men’s network, and 880.71, in the women’s network. This measure refers to how close or far a farmer is from all the other farmers in the network. Eigenvector quantifies the farmer’s level of influence in the network through the power and influence of their closest contacts. Therefore, farmers who present a high value on this measure are connected to many actors who are also relevant. On the contrary, farmers connected to other less-relevant actors will have a low eigenvector centrality. The results for this measure were low for both the men and the women, at 0.03 in both cases. 

Finally, the Bonacich Power was calculated. This measure refers to the power that an actor holds to be the one who connects his unpopular contacts since they depend on him. The results showed that the Bonacich Power for the men was 2179.66, while for the women, it was lower (1845.91).

According to the significant differences found in centrality measures for the reproductive area, we deepened the SNA in this area. Reproductive technologies were identified by Rangel et al. [13] and García et al. [17] in very small dual-purpose bovine farms from the Mexican tropics. This area was composed of the following seven technologies oriented to improve reproductive efficiency parameters: breeding soundness evaluation in bulls (T32), semen fertility evaluation (T33), evaluation of female body condition (T34), oestrus detection (T35), pregnancy diagnosis (T36), type of mating (seasonal or continuous) (T37), and breeding policy (T38). 

### 3.2. Technology Adoption Network Pattern of Dual-Purpose Farmers in the Reproduction Area

Figure 1 shows the dual-purpose farmer’s technological network in the reproduction area. The network consisted of 383 farmers, of which 91.12% were men and only 8.89% were women. To facilitate network visualization and interpretation, only farmers with an adoption rate higher than 57% in reproductive area were selected [46]. In the network, male farmers have been coloured green, while women have been coloured yellow. The network structure makes it possible to visualize how male and female farmers are interrelated. The network structure makes evident the existence of a core group composed of farmers f_1306, f_1501, f_824, f_510, f_442, f_175 and f_568. 

The men farmers located in the central part of the image reach the highest values of centrality, i.e., betweenness, acting as brokers, intermediating the communication that flows through the network (blue ellipse). This group was called leaders or pioneers in technological innovation.

In opposite, the men farmers located on the left side of the network present lower centrality values although they show great cohesion and homogeneity among themselves in the reproductive technology area (red ellipse). This group is called technological followers.

Regarding the role of women, the results showed that no women occupy central positions in the network and that only the farmer f_503 is highly integrated, close to other central farmers. The woman that appears in the image occupies an intermediate position between the technological leaders and followers, with higher centrality values (eigenvector) than the farmer on the left of the network. This woman reaches a strategic position, with closer connections to the technological leaders and is differentiated from the rest of the technological follower (violet ellipse).

The technology structure of the men’s adoption network is shown in Figure 2. In this case, the position of each technology is determined by its adoption frequency and by the centrality of the farmer who has adopted it.

The results showed that farmers have a strong preference for breeding soundness evaluation in bulls (T32), with an adoption rate of 96.85% (Table 5). This technology was considered basic. In addition, they tend to jointly adopt semen fertility evaluation (T33), evaluation of female body condition (T34), and pregnancy diagnosis (T36). Therefore, these were considered complementary technologies within the men’s adoption patterns.

On the contrary, the men’s adoption preferences positioned the technologies T35, T37 and T38 far from the core technologies. Following microeconomics principles, these were considered substitute technologies, with higher complexity and consequently a lower adoption rate. In microeconomics, two goods (technologies in this case) are substitutes if the products could be used for the same purpose by the consumers (farmers). That is, a farmer perceives both technologies as similar or comparable. Hence, having more of one technology causes the farmer to desire less of the other technology. Contrary to complementary technologies and basic or independent technologies, substitute technologies may replace each other in use due to changing farmers’ perceptions.

Figure 3 shows the technology structure of the adoption network for the women farmers. In this case, the structure was fuzzy. The women showed a strong preference for breeding soundness evaluation in bulls (T32), with an adoption rate of 94.12% (basic technology). In addition, they often jointly adopt the semen fertility evaluation (T33), pregnancy diagnosis (T36) and type of mating (T37) (complementary technologies). However, the women showed less preference for the evaluation of female body condition (T34), oestrus detection (T35) and breeding policy (T38) with a lower rate of adoption and far from each other in the network.

The comparison of reproductive technologies ratio between men and women is shown in Table 5. Significant differences in the type of mating (T37) have been found (*p* < 0.05).

### 3.3. Characterization of Small Farms According to Gender

The descriptive statistics for social, technical, and productive items of small dual-purpose farms in Mexico are shown in Table 6. The average age of the men was 51 years old, who generated one job on average, with a technical assistance rate of over two years. Livestock technical assistance refers to the years of technical support that a farmer has received from agricultural extension programs. The women turned out to be a little younger, with an average of 47 years, almost two jobs created and less technical assistance than the men (1.62 years). Statistically significant differences were found in the number of people who are economically dependent on the farm (*p* < 0.05). Regarding the educational level, only 2% of farmers had a medium level of education, most had basic education (76.2%), and 21.8% had no studies. In the case of the women, the majority had only basic education (61.8%), while 38.2% had no studies.

Regarding the productive performance variables of dual-purpose cattle farmers in Mexico, the results showed statistically significant differences in the values of milk production per ha, milk production per cow, and grazing area per ha (*p* < 0.05). 

By conducting a comparative analysis of men and women’s productive performances as shown in Table 7, the results showed that although women represent 8.89% of the population studied and had a lower average adoption rate than men, they were more productive in these variables. The women’s milk production was 121.37 L/ha, while the men showed an average milk production of 106.48 L/ha, which means that the men farmers produce almost 15 L/ha less than the women on average. Furthermore, considering milk production, the women showed better outcomes, reaching 11,781 L/year, while the annual production of the men was 11,217 L/year.

The women also showed better milk production outcomes per cow, with annual milk production of 1172 L/cow, while the men had an annual production of 971 L/cow, which means 201 L/cow less than the women. These results may be influenced by the fact that the men had almost three unproductive animals on average while the women had two. The men’s average grazing area was 28.09 ha, with a high coefficient of variation (143.43%), while the women showed a lower average grazing area than the men (19.29 ha).

## 4. Discussion

Interest in conceptualizing, measuring, and applying social network analysis (SNA) to gender differences in the rural world has grown tremendously in recent years [1,2,4,12,44,59]. While these studies have broadened our understanding of the role that social networks play in technology adoption in small scale farms, there have been less studies that have investigated the application of SNA to deepen understanding of how gender affects technology adoption in smallholders.

The development of gender studies in dual-purpose livestock in Mexico is novel and offers a different perspective to understand technology adoption. This study was adjusted to objective five of the Sustainable Development goals of the United Nations. Likewise, the application of social networks methodology is innovative and appropriate in this field of knowledge. The dual-purpose cattle system is the majority in Mexico and other Latin American countries, widespread in poor populations, with indigenous people, low levels of education, and where the role of women is invisible and not sufficiently valued [13,44,46].

According to Bullock et al. [60] and Contreras-Medina et al. [6], the gender perspective recognizes that some issues and constraints related to technological adoption success are gender-specific, and stem from the fact that both genders are different, have different needs, and face different constraints on several different levels. In this research, social networks analysis allowed us to identify the technology adoption network structure of Mexican dual-purpose farmers by gender, contributing to an understanding of the distinct roles and tasks that women and men assume, both within the family and household system and in relation to the community.

Regarding the position of men and women in the reproductive technology adoption network and their centrality measures, we have found that most technological brokers (leaders) are men. Likewise, male followers respond to a homogeneous and compact model with similar values of centrality among themselves. The women showed an intermediate position, closer to the brokers and far away from the group of male followers.

The results of this work were in line with those of [12], who studied the extent to which gender is a strong predictor of social ties in the chain value of rice from farmers in Benin, Niger, and Nigeria, and found that women occupy structurally unfavourable positions relative to men. The author argued that these gender disparities are particularly evident when actors are represented in terms of their capacity to play intermediary (broker) roles. Most of the brokers situated at the core of the network are men, and women rarely occupy structurally equivalent roles. However, the results of this work differ from those provided by [12] in terms of productive outcomes, since dual-purpose women, although they do not occupy central positions in the network, were well connected. This therefore allows them to access valuable information and be capable of better integrating it into the production process, achieving greater productivity. On the contrary, Walther et al. [12] found that farmers’ results were determined by their structural position within the rice value chain and, that the most prosperous actors were deeply embedded in their community through numerous links, being able to establish connections with other communities outside of their ethnic groups and countries.

SNA also made it possible to identify men’s and women’s preferences, adoption patterns and technological strategies. In the scope of the seven reproductive technologies evaluated, results showed that both men and women had strong preferences for breeding soundness evaluation in bulls (T32), with an adoption rate of 96.85%, although they differ in their adoption pattern both in complementary and substitutive technologies. As a result, we found that men tend to jointly adopt semen fertility evaluation (T33) and pregnancy diagnosis (T36). Therefore, these technologies are considered complementary technologies. On the one hand, evaluation of female body condition (T34) was the technology for which the men showed the least preference, with an adoption rate of 2.29%. On the other hand, the women tended to jointly adopt semen fertility evaluation (T33), pregnancy diagnosis (T36) and type of mating (T37). However, the women showed less preference for evaluation of female body condition (T34), and breeding policy (T38), with an adoption rate of 2.94% each. According to results found by Bullock et al. [60], women showed different patterns to exercise agency in economic and agricultural decision-making in Kenya. We have described patterns in gender relations and how innovation-specific decision-making may create spaces for women. Our key findings highlight how innovation processes often replicate gender patterns through decision-making in productive assets. However, access to agricultural knowledge offers avenues for women to expand their opportunity spaces by expanding their social networks and their ability to negotiate for resources in the household.

Besides, when conducting a comparative analysis of the productive performances of men and women, the results showed that although women represented 8.89% of the population studied, they showed higher levels in terms of milk productivity, and 20% of the differences in production per cow were explained by gender (971.26 L/cow in man, face to 1172.30 L/cow in women). Moreover, family structures were smaller in farms with women compared to men (1.74 vs. 1.80 economic dependents per farm). Finally, women’s farms showed smaller size of farm and, being smaller, they showed greater dairy specialization, with milk production being the core of the activity. In the discussion, only differences by gender have been considered, however within the group of men the results were different among brokers and follower men, i.e., f_175 is a broker with 2461 L/cow/year and 13 ha of grazing area, and conversely f_710 is follower with a production of 520 L/cow/year and 52 ha of grazing area. In summary, although the women were not the technological leaders, they quickly adopted technologies and surpassed the average results obtained by the men.

Despite the importance of gender differences in the process of innovation found, little is known about how men and women differ in technology adoption or farm management (risk aversion). Women tend to have greater tolerance for risk-taking than men in terms of willingness to try new or unusual products and enjoyment from the stimulation of newness. Women are more willing than men to adopt a new innovation earlier than other farmers [61,62]. Several researchers found that women (compared with men) are more open to uncertain and unstructured contexts. Conversely, Maio and Esses [63], Washburn, Smith, and Taglialatela, [64], and Maxfield et al. [65] found that men and women did not differ in their tendency for risk taking.

These results are in line with those of the project “Climate-smart livestock production and soil restoration in Uruguayan grasslands”, carried out by the Ministry of Livestock, Agriculture and Fisheries and Environment of Uruguay, where it was found that a livestock system with female participation in decision-making has a greater tendency to innovate and incorporate changes in the production system. The results also showed that there is a tendency for businesses run by women to have slightly higher productivity than the rest, with the higher educational level of women being one of the most differentiating characteristics between men and women [4].

The results of this work also differ from those found by Torres et al. [44], who studied the dual-purpose cattle system in Ecuador and found that women barely assume responsibilities in the production unit, with their work only being perceived as relevant in the domestic sphere. According to REDGATRO, and Rangel et al. [13], most dual-purpose farms have shown very low or null technical equipment, which is probably affecting the implementation of innovations. Therefore, the adoption of innovations could be associated with an improvement in technology and technical equipment. In this context, deepening the knowledge of technological patterns, and considering both basic and gender-specific technologies will help target these improvements.

This work favours a higher visibility of women’s activity and allows gender to be considered as a relevant factor in the results of the dual-purpose livestock system. In general, women, and socio-economic perspectives such as indigenous knowledge and people’s participation, have been largely ignored in energy planning and policy until recently [59]. 

According to Hay et al. [2], Rubio-Bañón et al. [66], and Contreras-Medina et al. [6], more research needs to be undertaken on how to incorporate differences in cultural organizations, traditional knowledge, and practices into the perspective of community empowerment.

Finally, it is important to point out that even when this research has some limitations, it also envisages future research lines. In this sense, the use of other technological indexes, such as the one developed by Gaworski et al. [67] to assess the technological progress level in the dairy production system, should be considered. There are also methodologies to consider, such as the structural equations with results of the farms [10].

## 5. Conclusions

The Social Network Analysis (SNA) was shown to be a useful tool to differentiate technology adoption patterns between men and women of dual-purpose cattle farms in tropical Mexico. In the centrality measurements of the reproductive technical area, significant differences between men and women were found. Men mostly occupied the central leadership positions of the network (brokers). 

Conversely, women occupied positions close to the leaders in the network, quickly adopting the reproductive technologies that they selected. The adoption pattern in basic technologies was similar between men and women. Differences were found in the adoption of complementary and substitute technologies between men and women. The women focused on those technologies linked to the reproductive efficiency of the cow.

Farms run by women were smaller (both in terms of cows and grazing area) although with greater specialization and milk productivity—both per cow and per ha, increasing productivity by 20%. Differences in adoption preferences between the men and the women and differences in technological combinations seemed to generate differences in productivity. The women achieved better outcomes with fewer technological resources.

In future studies, it would be necessary to delve into the reasons that cause women leaders not to appear in the network and their motivations for technological preferences.

## Figures and Tables

**Figure 1 animals-12-03194-f001:**
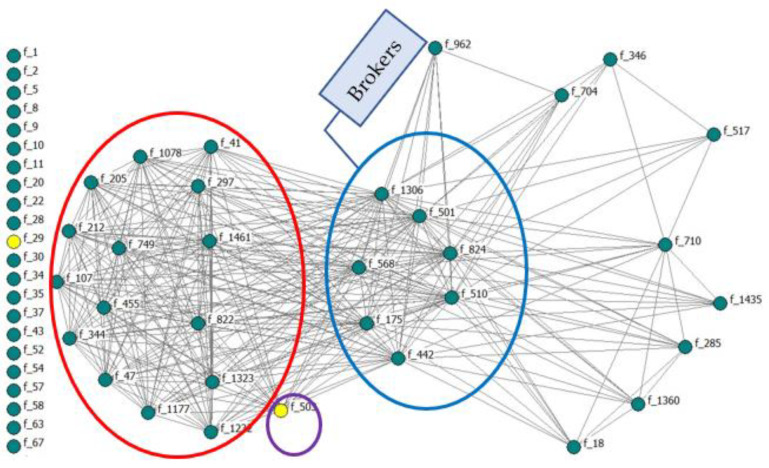
One-mode network visualization of farmers with an adoption higher than 57%. Farmer code (f_ni). Nodes’ colour by gender: 
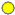
 Women, 
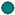
 Men.

**Figure 2 animals-12-03194-f002:**
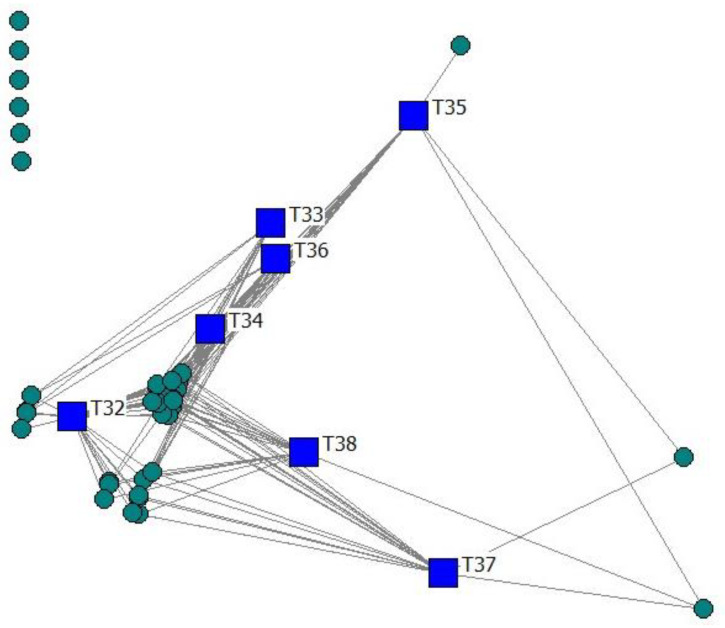
Two-mode network visualization of farmers and technologies. 
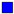
 Technologies, Farmer: 
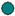
 Men.

**Figure 3 animals-12-03194-f003:**
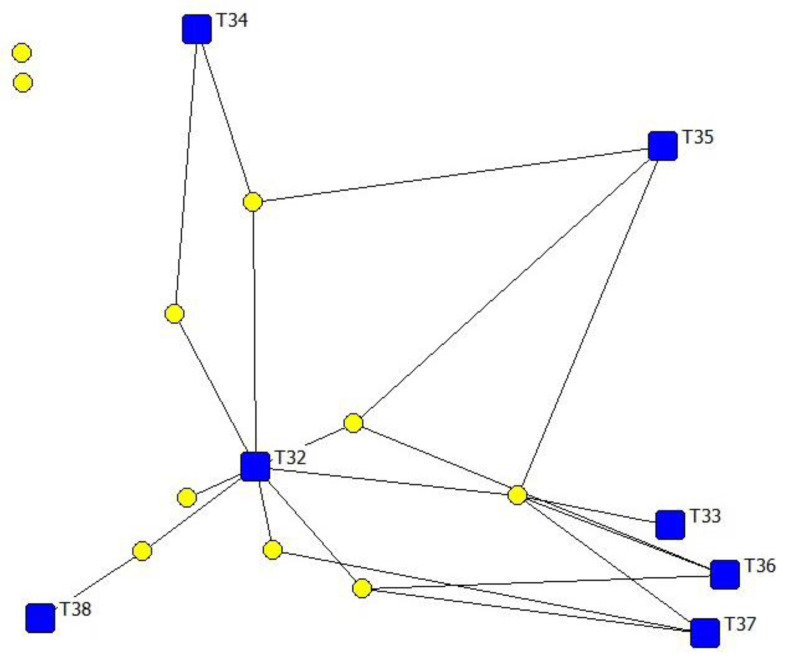
Two-mode network visualization of farmers and technologies. 
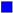
 Technologies, Farmer: 
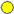
 Women.

**Table 1 animals-12-03194-t001:** Structural characteristics of dual-purpose cattle farms (n = 383) (Mean ± SD (CV, %) ^1^).

Indicator	Total	Minimum	Maximum
Age (years)	51.12 ± 14.51 (28.37)	20.00	85.00
Economic dependents (n)	2.91 ± 1.80 (62.01)	0.00	9.00
Employment, workers	1.50 ± 1.11 (74.28)	0.00	6.00
Technical assistance (years)	2.19 ± 1.99 (90.91)	0.00	11.00
Grazing area, ha	27.32 ± 38.96 (142.63)	3.00	400.00
Animal units, heads	19.32 ± 4.15 (21.47)	10.00	47.00
Stocking rate, AU/ha ^2^	1.09 ± 0.64 (58.29)	0.05	3.82
Herd size, nº cattle ^3^	25.64 ± 6.54 (25.54)	10.00	65.00
Milk production L/ha	107.80 ± 186.78 (173.26)	0.00	1428.57
Milk yield, L/year	11,267.24 ± 6807.19 (60.42)	0.00	36,500
Milk per cow, L/cow/year	989.11 ± 591.04 (59.75)	0.00	2940
Calves sold, nº calves ^4^	4.9 ± 5.8 (118.51)	0.00	40.00
Unproductive animals, heads	2.53 ± 4.52 (178.74)	0.00	32.00
Cheese yield, kg/farm/year	245.25 ± 733.71 (299.17)	0.00	9000
Total production cows	11.83 ± 2.88 (24.30)	3.00	22.00

^1^ CV: Coefficient of Variation; SD: Standard deviation; ^2^ AU/ha: Animal Unit per hectare; ^3^ Number of cattle; ^4^ Number of calves.

**Table 2 animals-12-03194-t002:** Descriptive statistics for technological areas (Mean ± SD (CV, %) ^1^).

Technological Packages (n: Technologies)	Group
Management (n = 8)	60.67 ± 10.64 (9.05)
Feeding (n = 14)	27.21 ± 6.29 (5.14)
Genetics (n = 9)	61.91 ± 10.60 (8.99)
Reproduction (n = 7)	15.45 ± 3.76 (9.21)
Animal Health (n = 7)	71.41 ± 14.72 (12.99)

^1^ CV: Coefficient of variation; SE: Standard deviation.

**Table 3 animals-12-03194-t003:** Descriptive statistics for technological packages (Mean ± SD (CV, %) ^1^).

Technological Packages	Men	Women	*p*-Value ^2^
Management	59.19 ± 6.61 (15.63)	60.66 ± 6.21 (15.74)	1.000 ns
Feeding	28.15 ± 6.32 (15.29)	25.84 ± 6.25 (14.51)	0.854 ns
Genetics	57.84 ± 6.52 (15.36)	57.84 ± 6.33 (14.86)	0.965 ns
Reproduction	29.41 ± 4.94 (11.82)	19.75 ± 4.32 (6.71)	0.109 ns
Animal Health	74.79 ± 5.18 (10.39)	70.17 ± 5.00 (9.31)	0.654 ns

^1^ CV: Coefficient of variation; SE: Standard deviation; ^2^
*p*-Value: ns = not significantly different between men and women.

**Table 4 animals-12-03194-t004:** Reproduction area technologies description (T, technology number).

Reproduction Technologies	Technologies Oriented to Improve Reproductive Efficiency Parameters
T32. Breeding soundness evaluation in bulls	Breeding soundness evaluation in bulls 0. No evaluation of the reproductive capacity of bulls or no sire on the farm; 1. Evaluation of the reproductive capacity of bull is carried out
T33. Semen fertility evaluation	Semen fertility evaluation 0. Sperm viability was not carried out; 1. Sperm fertility was evaluated
T34. Evaluation of female body condition	0. Evaluation of female body condition was not carried out; 1. Evaluation of female body condition was carried out before mating
T35. Estrus detection	0. Estrus detection was not carried out; 1. Estrus detection was carried out
T36. Pregnancy diagnosis	0. Pregnancy diagnosis was not carried out; 1. Pregnancy diagnosis was carried out as rectal palpation, ultrasound scanning, others
T37. Type of mating	0. Seasonal mating; 1. Continuous mating was carried out
T38. Breeding policy	0. Control of the mating was not carried out; 1. Planning mating control.

**Table 5 animals-12-03194-t005:** Centrality measures descriptive statistics for technological packages (Mean ± SD (CV, %) ^1^).

Technological Package		All	Men	Women	*p*-Value ^2^
Feeding	Degree	3.83 ± 2.26 (58.85)	3.85 ± 2.27 (58.91)	3.62 ± 2.11 (58.55)	0.545 ns
	Betweenness	36.02 ± 43.68 (121.25)	36.28 ± 44.46 (122.53)	33.36 ± 35.14 (121.25)	0.779 ns
	Closeness	1016 ± 247.25 (24.34)	1017.79 ± 251.29 (24.69)	997.56 ± 203.37 (20.39)	0.977 ns
	Eigenvector	0.032 ± 0.02 (51.10)	0.032 ± 0.02 (51.11)	0.032 ± 0.02 (51.75)	0.783 ns
	Bonacich Power	2632.59 ± 1345.39 (51.11)	2638.52 ± 1348.53 (51.11)	2571.78 ± 1330.99 (51.75)	0.781 ns
Reproduction	Degree	1.89 ± 1.37 (72.46)	1.94 ± 1.39 (71.79)	1.38 ± 0.99 (71.27)	0.010 *
	Betweenness	8.08 ± 14.51 (179.65)	8.42 ± 14.83 (176.18)	4.59 ± 10.25 (223.14)	0.029 *
	Closeness	841.19 ± 170.03 (20,21)	837.34 ± 156.99 (18.75)	**880.71 ± 270.13 (30.67)**	0.015 *
	Eigenvector	0.029 ± 0.01 (33.47)	0.03 ± 0.01 (33.02)	0.02 ± 0.07 (35.42)	0.012 *
	Bonacich Power	2150.03 ± 718.91 (33.44)	2179.66 ± 718.98 (32.99)	1845.91 ± 653.24 (35.39)	0.004 **
Health	Degree	5.08 ± 1.12 (22.07)	5.09 ± 1.127 (22.12)	4.91 ± 1.06 (21.48)	0.368 ns
	Eigenvector	0.036 ± 0.01 (16.98)	0.036 ± 0.01 (16.93)	0.035 ± 0.01 (17.77)	0.475 ns
	Closeness	793.65 ± 102.79 (12.95)	794.11 ± 107.35 (13.52)	789 ± 27.47 (3.48)	0.405 ns
	Betweenness	2.12 ± 2.15 (101.52)	2.17 ± 2.18 (100.57)	1.59 ± 1.745 (109.24)	0.248 ns
	Bonacich Power	4520.40 ± 767.63 (16.98)	4524.66 ± 765.91 (16.93)	4476.74 ± 795.49 (17.77)	0.459 ns
Genetics	Degree	5.33 ± 1.32 (24.87)	5.34 ± 1.34 (25.01)	5.21 ± 1.23 (23.54)	0.437 ns
	Betweenness	3.89 ± 3.00 (76.98)	3.93 ± 3.02 (76.88)	3.58 ± 2.81 (78.45)	0.297 ns
	Closeness	805.28 ± 954.13 (14.70)	806.46 ± 123.98 (15.37)	793.12 ± 4.09 (0.52)	0.485 ns
	Eigenvector	0.03 ± 0.01 (22.38)	0.04 ± 0.01 (22.85)	0.04 ± 0.01 (17.14)	0.481 ns
	Bonacich Power	4264.14 ± 954.12 (22.38)	4263.86 ± 973.94 (22.84)	4267.01 ± 731.42 (17.14)	0.475 ns
Management	Degree	4.85 ± 1.37 (28.20)	4.85 ± 1.37 (28.19)	4.85 ± 1.40 (28.76)	0.834 ns
	Betweenness	4.51 ± 6.68 (148.55)	4.61 ± 6.97 (151.26)	3.49 ± 1.95 (55.91)	0.799 ns
	Closeness	801.73 ± 103.72 (12.94)	802.45 ± 108.41 (13.51)	794.41 ± 23.39 (2.95)	0.851 ns
	Eigenvector	0.04 ± 0.01 (25.36)	0.04 ± 0.01 (25.39)	0.04 ± 0.01 (25.47)	0.697 ns
	Bonacich Power	4067.59 ± 1031.55 (25.36)	4065.43 ± 1032.04 (25.39)	4089.74 ± 1041.72 (25.47)	0.673 ns

^1^ CV: Coefficient of variation; SD: Standard deviation; ^2^
*p*-Value: * *p* < 0.05; ** *p* < 0.01; ns = not significantly different between men and women.

**Table 6 animals-12-03194-t006:** Technological adoption rate of men and women in reproductive area (T, technology number).

Code	Technology	Men	Women	*p*-Value ^1^
T32	Breeding soundness evaluation in bulls	96.85%	94.12%	0.403 ns
T33	Semen fertility evaluation	12.32%	2.94%	0.102 ns
T34	Evaluation of female body condition	2.29%	5.88%	0.211 ns
T35	Estrus detection	22.64%	8.82%	0.051 ns
T36	Pregnancy diagnosis	15.76%	11.76%	0.539 ns
T37	Type of mating	31.23%	11.76%	0.017
T38	Breeding policy	13.18%	2.94%	0.083 ns
Adoption rate in the reproduction area		27.75%	19.75%	

^1^*p* < 0.05; ns = not significantly different between men and women.

**Table 7 animals-12-03194-t007:** Technical and structural indicators by gender (Mean ± SD (CV, %) ^1^).

	Men	Women	*p*-Value ^2^
Age (years)	51.48 ± 14.26 (27.70)	47.47 ± 16.61 (35)	0.186 ns
Economic Dependents (n)	1.80 ± 1.79 (59.05)	1.74 ± 1.58 (91.18)	0.000 ***
Employments, workers	1.46 ± 1.11 (75.49)	1.79 ± 1.12 (62.55)	0.143 ns
Technical assistance (years)	2.25 ± 2.04 (90.69)	1.62 ± 1.33 (81.98)	0.129 ns
Milk production L/ha	106.48 ± 192.27 (180.57)	121.37 ± 117.43 (96.76)	0.007 **
Milk yield, L/year	11,217.13 ± 6815.17 (60.76)	11,781.65 ± 6804.18 (57.75)	0.529 ns
Milk per cow, L/cow/year	971.26 ± 580.04 (59.72)	1172.30 ± 676.64 (57.72)	0.049 *
Calves sold, nº calves ^3^	4.88 ± 5.79 (118.67)	5.12 ± 6.05 (118.24)	0.756 ns
Unproductive animals, heads	2.6 ± 4.63 (178.04)	1.82 ± 3.19 (175.55)	0.238 ns
Cheese yield, kg/farm/year	245.72 ± 735.55 (299.35)	240.32 ± 725.26 (301.79)	0.436 ns
Total production cows	11.92 ± 2.81 (23.58)	10.97 ± 3.39 (30.89)	0.940 ns
Animal units, heads	19.41 ± 4.18 (21.53)	18.33 ± 3.71 (20.27)	0.275 ns
Herd size, nº cattle ^4^	25.8 ± 6.61 (25.64)	24 ± 5.67 (23.62)	0.231 ns
Stocking rate, AU/ha ^5^	1.08 ± 0.62 (57.57)	1.19 ± 0.64 (64.11)	0.609 ns
Grazing area, ha	28.09 ± 40.29 (143.43)	19.29 ± 19.30 (100.02)	0.037 *

^1^ CV: Coefficient of variation; SE: Standard deviation; ^2^
*p*-Value: * *p* < 0.05; ** *p* < 0.01; *** *p* < 0.001; ns = not significantly different between men and women; ^3^ Number of calves; ^4^ Number of cattle; ^5^ AU: Animal unit per hectare.

## Data Availability

This is not applicable as the data are not in any data repository of public access, however if editorial committee needs access, we will happily provide them, please use this email: pa1gamaa@uco.es.

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
