# Peer review of "Does Gender Impact Technology Adoption in Dual-Purpose Cattle in Mexico?"

_animals, 2022, doi:10.3390/ani12223194_

Round 1
Reviewer 1 Report
The manuscript has scientific merits. The methods used are compatible with the objective. The results are presented clearly. However, despite the in-depth discussion of the results obtained, in the view of the present evaluator there are still gaps about the real explanation of the data. For what reason could such an explanation be asserted? Can there really be differences in technology adoption between men and women? Could the results be explained by factors prior to the activity (education, purchasing power, heritage, culture or others)? For example, from lines 53 to 57 there is no justification in the literature for the claim. Small adjustments would be necessary, such as the language in the note presented in Table 5. Despite being an interesting reading, in the view of the present evaluator, the manuscript does not include a convincing explanation of the results presented.
Author Response
We want to thank you for your work and valuable comments that have substantially help us to improve the quality of our initial manuscript. We have considered your comments and we have done all the suggested corrections.
Specifically, we have tried to clarify the indicated aspects.
*We have changed the footnotes of each table,
*An additional table (4) where the technologies are explained has been included
* We have changed some paragraphs throughout the manuscript and we have also reinforced the role of SNA in technological improvement.
as in line 170-177:
…Here, we focused on the more vulnerable and smaller group of commercial farmers with 20 or fewer cows, which represented 26% of the sample, distributed into dry and wet tropics. According to Rangel et al. [13], the 383 selected farms constituted a homogeneous cluster, small in size and with a low technological level which marketed most of the production. In addition, through the exploration of the data, we realized that the application of SNA is more significant and appropriate for groups with a lower technological level, since the differences can be better identified. Besides, being a group similar in economic dimension, the effect that it may have on technology adoption disappears….
We hope you like the new version of the manuscript.
Thank you very much for your attention.
Kind Regards,
The authors

Reviewer 2 Report
Regardless of the formulated general goal, it would also be worth writing in the article what was the cognitive (scientific) goal of the research and what utilitarian (useful) goals were set by the authors when undertaking research with a group of respondents? Moreover, the review of the state of knowledge in the Introduction provides the basis for the formulation of the research problem; one can write: The research problem was …. The research problem can also be associated with the indication of a gap in the current state of knowledge. Such issues of lack of knowledge were mentioned by the authors in the Introduction, they only need to be called a gap.
The authors used the acronym of the UW (line: 166), but its explanation is only in the further part of the text (line: 351), or actually in the explanations under Table 5. I suggest that the explanation of the UW should be given in the place where this acronym was used first once. By the way, I have a question: How is the UW interpreted in research? In the explanations under Tables 5 and 6, I found information that the UW is a Unit of work. However, I would like to ask what is the interpretation of "Unit of work". Are they man-hours, or man-hours per year, or man-hours per farm? I can guess that this is the standard number of hours worked in a year, but I am not sure about it and I do not know what the standard number of hours is in Thailand. Please complete this information in the article so that the reader would be able to clearly interpret the UW.
On line 165, the authors used the acronym UA and did not explain what that acronym means. I guess it's a usable area. But will all readers correctly interpret the acronym UA? It would be wise to give the full name of UA the first time you use UA.
I don't know if the phrase "... the yields of herd and land ..." (line: 167) is correct. In my opinion, it would be worth writing “annual capacity per cow” and “annual capacity per hectare”.
What does CV mean in the Tables titles? It is worth explaining this acronym so that the reader does not have to guess it, but be sure.
In Table 1, the superscript (1) was used, but there is no explanation for this notation under the Table. It would be worth supplementing this.
In Table 1, I do not understand the unit in "Stocking rate, UA / ha". In my interpretation, UA / ha would be Usable Area / ha. I do not understand this record. Or maybe the authors meant the number of animals (UA) per hectare. Just don't know how to elaborate this acronym in connection with animals; sure A means animals, but what is U? Please explain. Unfortunately, this example shows that there are many acronyms and other concepts in the article that may be incomprehensible to the reader.
Table 1 lists the term "Grazing Surface, ha". In my opinion, it would be more correct to say 'Grazing area, ha'. Surface in this case, however, probably has a different meaning and interpretation.
In Tables 1 and 6, as well as lines 361 and 453, the indicator "Economic dependents (n)" is mentioned. Unfortunately, I have nowhere found any information in the text of what the Authors understood by this concept. It would be worth providing more details about economic dependents so that you know what is going on and what this indicator covers, or it is calculated.
I have similar comments as above on the phrase "technical assistance" (included, for example, in Table 1). I know what technical assistance is, but specifying a unit (year) has shattered my certainty, what the Authors wanted to express by this term. Is it about the time of technical maintenance of farms by external companies or the time of using technical devices on the farm?
What was the criterion / criteria for selecting 383 farms for detailed research? Was it just the number of cows (herd size: 10-19 cows) or other factors?
Why was SE instead of SD included in the descriptive statistics of the data characterizing the research material? I mean the descriptive statistics of the data in Table 1 and other Tables.
I do not know what values are compared in Table 3, because I did not find information on this subject in the paragraph before the Table.
If the Authors use certain designations for technology, i.e. T32, T33, etc., these designations and their interpretations should be considered and described in the Materials and Methods chapter. In the article, it is possible to learn about the interpretation of the given markings only when the research results are presented, which causes a lot of confusion when reading the article. Therefore, I suggest that you explain the notations T32, T33, f_1306 etc. in the Materials and Methods chapter.
The implementation of technological solutions considered in the research and article involves the use of technical equipment. Was the technical equipment of dairy farms also included in the study? The technical equipment of the dairy production technology and its individual elements is also associated with the implementation of innovations; this concept was developed by the authors in the study. Therefore, it would be worth mentioning the technical equipment of the farms covered by the survey.
The article refers to the technology assessment in dairy production taking into account various, detailed areas of assessment included in this production. For the evaluation, the Authors used the technological innovation index, the technological level of adoption and others. I think that in the discussion of the research results, it would be worth paying attention to the fact that production technologies, including dairy production, can also be assessed using other indicators. An example of such an indicator (technological index level) was presented in the publication: Implementation of technical and technological progress in dairy production. It would be worthwhile to elaborate in the discussion or in the Introduction on issues showing progress in dairy production technologies, which are assessed in terms of who is responsible for the management of the dairy farm / farm with cattle.
The first sentence in the Conclusions chapter begins with the statement: The SNA methodology…. I had to go back to the beginning of the article to recall what SNA means. This is an example of the fact that it would be worth rewriting some sentences in an article in such a way that they would be more understandable for the reader, or the reader does not have to refer to the previous paragraphs and look for the meaning of some acronyms.
Author Response
We want to thank you for your work and valuable comments that have substantially help us to improve the quality of our initial manuscript. We have considered your comments and we have done all the suggested corrections.
Please, find our answer to each of your suggestions in the description below.
We hope you like the new version of the manuscript.
Thank you very much for your attention.
Kind Regards,
The authors
Comments and Suggestions for Authors
- Regardless of the formulated general goal, it would also be worth writing in the article what was the cognitive (scientific) goal of the research and what utilitarian (useful) goals were set by the authors when undertaking research with a group of respondents? Moreover, the review of the state of knowledge in the Introduction provides the basis for the formulation of the research problem; one can write: The research problem was …. The research problem can also be associated with the indication of a gap in the current state of knowledge. Such issues of lack of knowledge were mentioned by the authors in the Introduction, they only need to be called a gap.
According to your suggestions, we have tried to clarify the indicated aspects and included the scientific objectives as follow:
This study contributes to deepening the knowledge of the low technology adoption rate in DP livestock systems from the gender perspective, as most of the studies in the field have estimated the network’s structure based on adoption intentions [6,11]. On the opposite, this research estimated networks from the reproductive technologies already adopted by farmers, identifying the actual technological adoption patterns by gender. According to De-Pablos-Heredero et al. [10] and Walther et al. [12], male and female farmers with different technological patterns and betweenness power in the network will achieve different productive results…
… The dual-purpose livestock systems in the Mexican tropics respond to small-scale production, and are key to the food security of the inhabitants of the tropics; both in terms of provision and access to food, stability, and prices [8]. Globally, dual-purpose systems generate between 19% and 12% of meat and milk world’s production [9,10] and live on the poverty line, in fragile extensive systems with a high degree of marginalization. In addition, these subsistence farms showed a low level of technological innovation, which makes it difficult to access external inputs and are highly vulnerable to environmental disasters and economic turmoil [11]. In this context, the research problem was the gap in the current state of knowledge about the reasons for the low technology adoption rate in this livestock system and the grounds why some farmers adopt or reject technologies potentially available to them…
- The authors used the acronym of the UW (line: 166), but its explanation is only in the further part of the text (line: 351), or actually in the explanations under Table 5. I suggest that the explanation of the UW should be given in the place where this acronym was used first once. By the way, I have a question: How is the UW interpreted in research? In the explanations under Tables 5 and 6, I found information that the UW is a Unit of work. However, I would like to ask what is the interpretation of "Unit of work". Are they man-hours, or man-hours per year, or man-hours per farm? I can guess that this is the standard number of hours worked in a year, but I am not sure about it and I do not know what the standard number of hours is in Thailand. Please complete this information in the article so that the reader would be able to clearly interpret the UW.
We have given the explanation of the UW in the place where this acronym was used first, and we have added the interpretation of Unit of work.
- On line 165, the authors used the acronym UA and did not explain what that acronym means. I guess it's a usable area. But will all readers correctly interpret the acronym UA? It would be wise to give the full name of UA the first time you use UA.
This was an error from the Spanish translation, we have added the correct acronym and its meaning of Animal units (AU), as follows:
The main characteristics of this group are shown in table 1. These producers exhibited an average age of 51.12 years old, the youngest farmer being 20 years of age, and the oldest 85 years of age, with nearly three people directly depending on the farms, generating 1.5 working units and 2.19 years of experience in technical advice. These were low production and dimension farms. The average size was 27.32 ha, 11.83 cows and 19.32 animal units (AU).
- I don't know if the phrase "... the yields of herd and land ..." (line: 167) is correct. In my opinion, it would be worth writing “annual capacity per cow” and “annual capacity per hectare”.
We have changed “… the yields of herd and land …” for “…annual capacity per cow and hectare.
- What does CV mean in the Tables titles? It is worth explaining this acronym so that the reader does not have to guess it, but be sure.
We have reviewed each table and its acronyms have been corrected so that it is clearer for the reader, and we have added the explanation for the acronym CV, which means coefficient of variation.
- In Table 1, the superscript (1) was used, but there is no explanation for this notation under the Table. It would be worth supplementing this.
As we indicated before, we have reviewed each table and its acronyms have been corrected, and we have added the explanation for the acronym CV, which means coefficient of variation as follows:
Table 1. Structural characteristics of dual‐purpose cattle farms (n = 383) (Mean ± SD (CV, %)1)
|
Indicator |
Total |
Minimum |
Maximum |
|
Age (years) |
51.12 ± 14.51 (28.37) |
20.00 |
85.00 |
|
Economic dependents (n) |
2.91 ± 1.80 (62.01) |
0.00 |
9.00 |
|
Employments, workers |
1.50 ± 1.11 (74.28) |
0.00 |
6.00 |
|
Technical assistance (years) |
2.19 ± 1.99 (90.91) |
0.00 |
11.00 |
|
Grazing area, ha |
27.32 ± 38.96 (142.63) |
3.00 |
400.00 |
|
Animal units, heads |
19.32 ± 4.15 (21.47) |
10.00 |
47.00 |
|
Stocking rate, AU/ ha2 |
1.09 ± 0.64 (58.29) |
0.05 |
3.82 |
|
Herd size, nº cattle |
25.64 ± 6.54 (25.54) |
10.00 |
65.00 |
|
Milk production l/ha |
107.80 ± 186.78 (173.26) |
0.00 |
1,428.57 |
|
Milk yield, l/year |
11,267.24 ± 6,807.19 (60.42) |
0.00 |
36,500 |
|
Milk per cow, l/cow/year |
989.11 ± 591.04 (59.75) |
0.00 |
2,940 |
|
Calves sold, nº calves |
4.9 ± 5.8 (118.51) |
0.00 |
40.00 |
|
Unproductive animals, heads |
2.53 ± 4.52 (178.74) |
0.00 |
32.00 |
|
Cheese yield, kg/farm/year |
245.25 ± 733.71 (299.17) |
0.00 |
9,000 |
|
Total production cows |
11.83 ± 2.88 (24.30) |
3.00 |
22.00 |
1CV: Coefficient of Variation; SD: Standard deviation; 2AU: Animal Unit per hectare
- In Table 1, I do not understand the unit in "Stocking rate, UA / ha". In my interpretation, UA / ha would be Usable Area / ha. I do not understand this record. Or maybe the authors meant the number of animals (UA) per hectare. Just don't know how to elaborate this acronym in connection with animals; sure, A means animals, but what is U? Please explain. Unfortunately, this example shows that there are many acronyms and other concepts in the article that may be incomprehensible to the reader.
As we have mentioned before, this was an error from the Spanish translation, we have added the correct acronym and its meaning, which is Animal Units (AU). We have corrected all the acronym mistakes.
- Table 1 lists the term "Grazing Surface, ha". In my opinion, it would be more correct to say 'Grazing area, ha'. Surface in this case, however, probably has a different meaning and interpretation.
We have replaced “Grazing Surface” by Grazing area when required throughout the manuscript.
- In Tables 1 and 6, as well as lines 361 and 453, the indicator "Economic dependents (n)" is mentioned. Unfortunately, I have nowhere found any information in the text of what the Authors understood by this concept. It would be worth providing more details about economic dependents so that you know what is going on and what this indicator covers, or it is calculated.
We have corrected the indicator Economic dependants and added its meaning, which is the number of people economic dependent of the farm.
The women turned out to be a little younger, with an average of 47 years, almost two jobs created and less technical assistance than the men (1.62 years). Statistically significant differences were found in the number of people economic dependent of the farm (p < 0.05).
- I have similar comments as above on the phrase "technical assistance" (included, for example, in Table 1). I know what technical assistance is, but specifying a unit (year) has shattered my certainty, what the Authors wanted to express by this term. Is it about the time of technical maintenance of farms by external companies or the time of using technical devices on the farm?
We have added the definition of technical assistance as follow:
…The descriptive statistics for social, technical, and productive items of small dual‐purpose farms in Mexico are shown in Table 6. The average age of the men was 51 years old, who generated one job on average, with a technical assistance rate of over two years. Livestock technical assistance refers to the years of technical support that a farmer has received from agricultural extension programs…
- What was the criterion / criteria for selecting 383 farms for detailed research? Was it just the number of cows (herd size: 10-19 cows) or other factors?
We explain a little bit more in the manuscript why we have chosen this group:
…Here, we focused on the more vulnerable and smaller group of commercial farmers with 20 or fewer cows, which represented 26% of the sample, distributed into dry and wet tropics. According to Rangel et al. [13], the 383 selected farms constituted a homogeneous cluster, small in size and with a low technological level which marketed most of the production. In addition, through the exploration of the data, we realized that the application of SNA is more significant and appropriate for groups with a lower technological level, since the differences can be better identified. Besides, being a group similar in economic dimension, the effect that it may have on technology adoption disappears…
- Why was SE instead of SD included in the descriptive statistics of the data characterizing the research material? I mean the descriptive statistics of the data in Table 1 and other Tables.
We have changed the SE by SD (Standard Deviation)
- I do not know what values are compared in Table 3, because I did not find information on this subject in the paragraph before the Table.
We have explained what is compared in table 3 as follow:
Table 3 shows the values of Mean, standard deviation (SD) and the coefficient of variation (CV) for the five technological packages studied: Management, Feeding, Genetics, Reproduction and Animal Health. The five technological areas showed similar values between men and women, in addition no significant differences were found due to the gender factor (p>0.05).
- If the Authors use certain designations for technology, i.e. T32, T33, etc., these designations and their interpretations should be considered and described in the Materials and Methods chapter. In the article, it is possible to learn about the interpretation of the given markings only when the research results are presented, which causes a lot of confusion when reading the article. Therefore, I suggest that you explain the notations T32, T33, f_1306 etc. in the Materials and Methods chapter.
We have added de technologies description in the Materials and Methods chapter, and explained the notations as follows:
Table 4. Reproduction area technologies description
|
Reproduction technologies |
Technologies oriented to improve reproductive efficiency parameters |
|
T32. Breeding soundness evaluation in bulls |
Breeding soundness evaluation in bulls |
|
T33. Semen fertility evaluation |
Semen fertility evaluation |
|
T34. Evaluation of female body condition |
0. Evaluation of female body condition was not done; 1. Evaluation of female body condition was done before mating |
|
T35. Estrus detection |
0. Estrus detection was not done; 1. Estrus detection was done |
|
T36. Pregnancy diagnosis |
0. Pregnancy diagnosis was not done; 1. Pregnancy diagnosis was done as rectal palpation, ultrasound scanning, others |
|
T37. Type of mating |
0. Seasonal mating; 1. Continuous mating was done |
|
T38. Breeding policy |
0. Control of the mating was not done; 1. Planning mating control. |
- The implementation of technological solutions considered in the research and article involves the use of technical equipment. Was the technical equipment of dairy farms also included in the study? The technical equipment of the dairy production technology and its individual elements is also associated with the implementation of innovations; this concept was developed by the authors in the study. Therefore, it would be worth mentioning the technical equipment of the farms covered by the survey.
We have mentioned the technical equipment of the farms as shown below:
The results of this work also differ from those found by Torres et al. [44], who studied the dual-purpose cattle system in Ecuador and found that women barely assume responsibilities in the production unit, being their work perceived as relevant only in the domestic sphere. According to REDGATRO, and Rangel et al. [13], most dual-purpose farms have shown very low or null technical equipment, which is probably affecting the implementation of innovations. Therefore, the adoption of innovations could be associated with an improvement in technology and technical equipment. In this context, deepening the knowledge of technological patterns, and considering both basic and gender-specific technologies will help target these improvements.
- The article refers to the technology assessment in dairy production taking into account various, detailed areas of assessment included in this production. For the evaluation, the Authors used the technological innovation index, the technological level of adoption and others. I think that in the discussion of the research results, it would be worth paying attention to the fact that production technologies, including dairy production, can also be assessed using other indicators. An example of such an indicator (technological index level) was presented in the publication: Implementation of technical and technological progress in dairy production. It would be worthwhile to elaborate in the discussion or in the Introduction on issues showing progress in dairy production technologies, which are assessed in terms of who is responsible for the management of the dairy farm / farm with cattle.
We have carefully considered your suggestions and have incorporated them as limitations and future lines of work.
Finally, it is important to point out that even when this research has some limitations, it also envisages future research lines. In this sense, the use of other technological indexes as the one developed by Gaworski et al. [67], to assess the technological progress level in the dairy production system should be considered. There are also methodologies to consider, such as the structural equations with results of the farms [10]
- The first sentence in the Conclusions chapter begins with the statement: The SNA methodology…. I had to go back to the beginning of the article to recall what SNA means. This is an example of the fact that it would be worth rewriting some sentences in an article in such a way that they would be more understandable for the reader, or the reader does not have to refer to the previous paragraphs and look for the meaning of some acronyms.
We have revised the entire manuscript, clarified the sentences and rewritten the first sentence in the conclusion as follows:
- Conclusions
The Social Network Analysis (SNA) was shown to be a useful tool to differentiate technology adoption patterns between men and women of dual-purpose cattle farms in tropical Mexican. In the centrality measurements of the reproductive technical area, significant differences between men and women were found. Men mostly occupied the central leadership positions of the network (brokers).

Reviewer 3 Report
This manuscript can be accepted after minor revision.
1. Line 67: 19%
2. Line 76: not only in Latin America, please extend this phrase.
3. Line 136: the statement “in developing countries and” should be deleted.
4. Line 162: About the age please give Median (Min-Max) statistics which are more informative for this variable (also in Table 1).
5. Table 1: “Animal units, heads”, “Herd size, nº cattle”, “Calves sold, nº calves”, and “Unproductive animals, heads” should be given as Median (Min-Max) statistics.
6. In Table 3, 4, 5 and 6: “p-Value1” superscript 1 is not defined anywhere or meaningless.
7. Line 317: “figure” should be “Figure”
8. Table 5: 0.051 of P value is not statistically significant.
9. Table 5: “Tasa de adopción en el área de reproducción” what does it mean?

Author Response
We want to thank you for your work and valuable comments that have substantially help us to improve the quality of our initial manuscript. We have considered your comments and we have done all the suggested corrections.
Please, find our answer to each of your suggestions in the description below.
We hope you like the new version of the manuscript.
Thank you very much for your attention.
Kind Regards,
The authors
- Line 67: 19%
We have corrected this, adding the symbol (%) as it's shown in the following:
The dual-purpose livestock systems in the Mexican tropics respond to small-scale production, and are key to the food security of the inhabitants of the tropics; both in terms of provision and access to food, stability, and prices [8]. Globally, dual-purpose systems generate between 19% and 12% of meat and milk world’s production [9,10] and live on the poverty line, in fragile extensive systems with a high degree of marginalization.
- Line 76: not only in Latin America, please extend this
Case studies from other countries such as India, Kenya, etc. have been incorporated as it's shown in the following:
Historically, the low technology adoption rate has been one of the main problems of small-scale livestock systems in Latin America and other developing countries [1,3,4,10,11,12], which negatively influences their productivity and threatens the food security of many families whose livelihood depends on livestock farming [10,13].
- Line 136: the statement “in developing countries and” should be deleted.
We have deleted the statement.
- Line 162: About the age, please give Median (Min-Max) statistics which are more informative for this variable (also in Table 1).
We have added the minimum and maximum age in the text and in table 1, as follow:
Table 1. Structural characteristics of dual‐purpose cattle farms (n = 383) (Mean ± SD (CV, %)1)
|
Indicator |
Total |
Minimum |
Maximum |
|
Age (years) |
51.12 ± 14.51 (28.37) |
20.00 |
85.00 |
|
Economic dependents (n) |
2.91 ± 1.80 (62.01) |
0.00 |
9.00 |
|
Employments, workers |
1.50 ± 1.11 (74.28) |
0.00 |
6.00 |
|
Technical assistance (years) |
2.19 ± 1.99 (90.91) |
0.00 |
11.00 |
|
Grazing area, ha |
27.32 ± 38.96 (142.63) |
3.00 |
400.00 |
|
Animal units, heads |
19.32 ± 4.15 (21.47) |
10.00 |
47.00 |
|
Stocking rate, AU/ ha2 |
1.09 ± 0.64 (58.29) |
0.05 |
3.82 |
|
Herd size, nº cattle |
25.64 ± 6.54 (25.54) |
10.00 |
65.00 |
|
Milk production l/ha |
107.80 ± 186.78 (173.26) |
0.00 |
1,428.57 |
|
Milk yield, l/year |
11,267.24 ± 6,807.19 (60.42) |
0.00 |
36,500 |
|
Milk per cow, l/cow/year |
989.11 ± 591.04 (59.75) |
0.00 |
2,940 |
|
Calves sold, nº calves |
4.9 ± 5.8 (118.51) |
0.00 |
40.00 |
|
Unproductive animals, heads |
2.53 ± 4.52 (178.74) |
0.00 |
32.00 |
|
Cheese yield, kg/farm/year |
245.25 ± 733.71 (299.17) |
0.00 |
9,000 |
|
Total production cows |
11.83 ± 2.88 (24.30) |
3.00 |
22.00 |
1CV: Coefficient of variation; SD: Standard deviation; 2AU: Animal Unit per hectare
- Table 1: “Animal units, heads”, “Herd size, nº cattle”, “Calves sold, nº calves”, and “Unproductive animals, heads” should be given as Median (Min-Max) statistics.
We have modified table 1 and added the minimum and maximum values to each item as have been shown before in the comment number 4.
- In Table 3, 4, 5 and 6: “p-Value1” superscript 1 is not defined anywhere or meaningless.
We have reviewed each of the tables, all the abbreviations and footnotes in order to make them homogeneous.
- Line 317: “figure” should be “Figure”
We have changed it to capital letter as follow:
The technology structure of the men's adoption network is shown in Figure 2. In this case, the position of each technology is determined by its adoption frequency and by the farmer's centrality who has adopted it.
- Table 5: 0.051 of P value is not statistically significant.
We have changed it both, in the table and in the text as follow:
- Table 5: “Tasa de adopción en el área de reproducción” what does it mean?
It has been an error from the Spanish translation. We have changed it as follow:
Table 6. Technological adoption rate of men and women in reproductive area
|
Code |
Technology |
Men |
Women |
p-Value1 |
|
T32 |
Breeding soundness evaluation in bulls |
96,85% |
94,12% |
0.403 ns |
|
T33 |
Semen fertility evaluation |
12,32% |
2,94% |
0.102 ns |
|
T34 |
Evaluation of female body condition |
2,29% |
5,88% |
0.211 ns |
|
T35 |
Estrus detection |
22,64% |
8,82% |
0.051 ns |
|
T36 |
Pregnancy diagnosis |
15,76% |
11,76% |
0.539 ns |
|
T37 |
Type of mating |
31,23% |
11,76% |
0.017 |
|
T38 |
Breeding policy |
13,18% |
2,94% |
0.083 ns |
|
Adoption rate in the reproduction area |
27,75% |
19,75% |
|
|
1p < 0.05; ns = not significantly different between men and women.

Round 2
Reviewer 1 Report
The authors made improvements in the quality of the material. In the view of this reviewer, the manuscript has merit to be published.
Reviewer 2 Report
Thank you for the changes and additions made to the article, as well as for taking into account the suggestions presented in the review.